# pH-Dependent Molecular Gate Mesoporous Microparticles for Biological Control of *Giardia intestinalis*

**DOI:** 10.3390/pharmaceutics13010094

**Published:** 2021-01-13

**Authors:** Isabel González-Alvarez, Verónica Vivancos, Carmen Coll, Bárbara Sánchez-Dengra, Elena Aznar, Alejandro Ruiz-Picazo, Marival Bermejo, Félix Sancenón, María Auxiliadora Dea-Ayuela, Marta Gonzalez-Alvarez, Ramón Martínez-Máñez

**Affiliations:** 1Department of Engineering, Pharmacokinetics and Pharmaceutical Technology Area, Miguel Hernandez University, Elche. San Juan Campus, 03550 San Juan, Spain; isabel.gonzalez@umh.es (I.G.-A.); cucunica@hotmail.com (V.V.); barbarasanchezdengra@gmail.com (B.S.-D.); alejandroruizpicazo@gmail.com (A.R.-P.); mbermejo@goumh.umh.es (M.B.); 2Instituto Interuniversitario de Investigación de Reconocimiento Molecular y Desarrollo Tecnológico, Universitat Politècnica de València and Universitat de València, Camino de Vera s/n, 46022 Valencia, Spain; carmencollmerino@gmail.com (C.C.); elazgi@upvnet.upv.es (E.A.); fsanceno@upvnet.upv.es (F.S.); rmaez@qim.upv.es (R.M.-M.); 3CIBER de Bioingeniería, Biomateriales y Nanomedicina (CIBER-BBN), 46022 Valencia, Spain; 4Unidad Mixta UPV-CIPF de Investigación en Mecanismos de Enfermedades y Nanomedicina, Universitat Politècnica de València, Centro de Investigación Príncipe Felipe, C/Eduardo Primo Yúfera 3, 46012 Valencia, Spain; 5Unidad Mixta de Investigación en Nanomedicina y Sensores, Universitat Politècnica de València, IIS La Fe, Avenida Fernando Abril Martorell 106, 46026 Valencia, Spain; 6Departamento de Farmacia, Facultad de Ciencias de la Salud, Universidad Cardenal Herrera-CEU, C/Santiago Ramón y Cajal, s/n, Alfara del Patriarca, 46115 Valencia, Spain; mda_3000@yahoo.es

**Keywords:** mesoporous microparticles, *G. intestinalis*, molecular gate, targeted drug delivery, oral administration

## Abstract

Giardiasis is a parasitism produced by the protozoa *Giardia intestinalis* that lives as trophozoite in the small intestine (mainly in the duodenum) attached to the intestinal villus by means of billed discs. The first line treatment is metronidazole, a drug with high bioavailability, which is why to obtain therapeutic concentrations in duodenum, it is necessary to administer high doses of drug to patients with the consequent occurrence of side effects. It is necessary to developed new therapeutical approaches to achieve a local delivery of the drug. In this sense, we have developed gated mesoporous silica microparticles loaded with metronidazole and with a molecular gate pH dependent. In vitro assays demonstrated that the metronidazole release is practically insignificant at acidic pHs, but in duodenum conditions, the metronidazole delivery from the microparticles is effective enough to produce an important parasite destruction. In vivo assays indicate that this microparticulate system allows to increase the concentration of the drug in duodenum and reduce the concentration in plasma avoiding systemic effects. This system could be useful for other intestinal local treatments in order to reduce doses and increase drug availability in target tissues.

## 1. Introduction

Giardiasis is a parasitic diarrheal disease caused by the parasite *Giardia intestinalis* (also known as *Giardia lamblia* or *Giardia duodenalis*), a flagellated protozoan which is able to colonize several animals’ small intestine, including the human one [1]. Symptoms may include diarrhea, abdominal pain, and weight loss and, less frequently, vomiting, blood in the stool, and fever. Giardiasis is one of the most common parasitic diseases globally and it is included in the The World Health Organization (WHO) Neglected Diseases Initiative since 2004 [2], as it affects children more harshly, even causing them severe malnutrition, delayed physical development and poor cognitive function [3]. In addition, according with epidemiological data, giardiasis is present in nearly 33% of the population of developing countries and in nearly 2% of adults and 6–8% of children of developed countries [4]. Transmission of the disease occurs with the ingestion of cysts by fecal–oral route. Furthermore, giardiasis spreads easily through a population, because one infected person can produce about 1–10 billion cysts daily, while just swallowing 10 cysts can cause an illness [3].

Regarding giardiasis treatment, it is difficult to compare studies which evaluate efficacy of different drugs, due to the methodological differences between investigations. However, there is no ideal treatment for giardiasis that allows high levels of success without adverse (or limited) effects [5]. Currently, metronidazole is considered the first-line treatment against giardiasis [6], mainly, because of its high efficacy which is between 80 and 95%. Metronidazole is a derivative of 5-nitroimidazole whose mechanism of killing *Giardia* has been studied in detail. 5-nitroimidazoles derivatives, as metronidazole, penetrate inside the microorganism by passive diffusion. Once there, the drugs are metabolized by nitroreductases, enzymes that transform, by reduction, the nitro group of the drug structure to an amine, generating different intermediate oxidative species as free radicals. These intermediate metabolites are able to denaturalize microbial DNA, provoking the loss of the helical structure and cell death [7].

Nevertheless, the treatment with metronidazole has a number of drawbacks which make it, even being the drug of choice, not the ideal drug. Some of these disadvantages are the emergence of a 10% of resistant strains [8,9]; the bad taste of the compound that hinders the dosage in children; and its most common adverse effects that include dizziness, metallic taste, nausea, diarrhea, abdominal pain, anorexia, respiratory infection, and flu-like symptoms [7]. In addition, probably the most important drawback in the use of metronidazole, is its very high (nearly 100%) bioavailability, which gives it a high absorption in the digestive tract [5]. This high bioavailability means a high absorption and low levels of metronidazole reaching the duodenum, the site of action where *Giardia* is accumulated. So, to reach therapeutic drug levels against *Giardia* in small intestine, it is necessary to increase the dose, which leads to a rise in the gravity and number of systemic adverse reactions.

Taking into account all of the above, the search of new specific targeted formulations or new treatment options are of importance to overcome these limitations. A suitable option is to encapsulate the drug in smart nano or microdevices to target duodenum specifically. For this aim, particular conditions of gastrointestinal tract, such as transit time, pH changes, or specific enzymes or molecules, has to be taken into account [10,11,12,13]. In fact, during the last few years, the development of microdevices able to deliver their cargo under certain specific conditions has been extensively explored [14].

One of the more effective strategies to get release just in one section of the intestine, such as the duodenum, is to use pH-sensitive nano or microdevices [15,16]. In the scientific literature, a wide variety of scaffolds, such as polymeric supports, made, for instance, with polyethylene glycol (PEG), chitosan, hyaluronic acid or albumin; micellar supports, with, for example, PEG, polycaprolactone, SPION and folate in their structure; liposomal supports or mesoporous silica supports [17], have been functionalized with different molecules to get pH-responsive nanocarriers. Among them, mesoporous silica have demonstrated great potential as delivery systems due to their excellent physicochemical properties: biocompatibility chemical inertness, sturdiness, thermal stability, homogenous porosity, high load capacity, and ease of functionalization [18,19]. Moreover, mesoporous silica can be equipped with “molecular gates” (also known as gate keepers or nanovalves) allowing the preparation of materials showing “zero” release, yet being able to deliver the payload on-command using external stimuli. In fact, gated materials have been used to develop different applications such as drug controlled release [20,21,22], sensing [23], or advanced communication protocols [24,25].

Molecular gates engineered by incorporating polyamines into the pore outlets of a silica support loaded with the drug, has been reported a suitable strategy to get controlled release at certain pHs. In polyamine-functionalized mesoporous silica amine groups are protonated at acidic pH and, therefore, they repel each other, causing pore blockage. Furthermore, the protonated amines strongly interact electrostatically with anions, resulting in a better pore closure. Conversely, at higher pH amine groups are less protonated, repulsion between amines and interaction with anions decrease thus causing pore opening and cargo delivery [26]. This pH dependent behavior might be applied to achieve specific delivery in body areas with different pHs as intestinal tract, vaginal cavity, or tumor surroundings.

Based on the above, we report herein the preparation of mesoporous silica microparticles loaded with metronidazole and capped with polyamines able to respond to pH changes, in order to obtain drug delivery specifically in the duodenum, reducing thereby the previous drug delivery and consequently the dose needed to treat the disease and the systemic adverse effects. Research efforts in this area are required to obtain more safe and effective treatments to improve specially children health in developing countries.

## 2. Materials and Methods

### 2.1. Reagents and Solvents

For the synthesis of the microparticulated mesoporous material, tetraethylorthosilicate (TEOS), cetyltrimethylammonium bromide (CTABr) and triethanolamine (TEAH_3_) were supplied by Aldrich. N′-(3-trimethoxysilyl) propyl diethylenetriamine used for surface functionalization of the mesoporous materials, safranin O (to set up the system) and metronidazole used for the loading of the microparticles were also acquired from Aldrich. Sodium hydroxide (NaOH) was purchased from Scharlab. All products were used without any purification method.

### 2.2. Synthesis of MCM-41 Microparticulated Solid

MCM-41 mesoporous support was obtained following the atrane route [27]. Molar ratio between the different reagents used in the synthesis was 7 TEAH_3_:2 TEOS:0.52 CTABr:0.5 NaOH:180 H_2_O. In a typical synthesis, 0.98 g of NaOH (previously dissolved in 2 mL of H_2_O) were added to 52.4 g of triethanolamine under constant stirring. The mixture was heated until 120 °C to generate the atrane complexes. Then, temperature was set at 70 °C and 22 mL of TEOS were added. In a further step, the mixture was heated to 118 °C and the structure-directing agent CTABr (9.26 g) was slowly added. Once it is dissolved, the mixture temperature was set at 70 °C and 180 mL of H_2_O were added to promote the hydrolysis of the silica precursor.

Few minutes later, the formation of a whitish suspension was observed and maintained at room temperature for one hour and then placed in a Teflon autoclave to be aged again for 24 h at 100 °C. The obtained material was centrifuged and washed with water until reach neutral pH. In the last step, the obtained mesostructured MCM-41 material was dried at 70 °C for 24 h. Subsequently, the surfactant was eliminated by calcination in the presence of air at 550 °C and the final material (MCM-41) with an ordered mesoporous structure was obtained.

#### 2.2.1. MCM-Met-N3 Solid Preparation

To obtain the drug loaded material, a slightly modified protocol was used [26]. 68.5 mg (0.4 mmol) of metronidazole were completely dissolved in 15 mL of acetonitrile. Then, 0.5 g of calcined MCM-41 were added, and the mixture was stirred for 24 h to allow drug diffusion into the pores. In a further step, an excess of N′-(3-trimethoxysilyl) propyl diethylenetriamine (2.5 mL, 7.5 mmol) was added to the mixture and the suspension was left under stirring for 6 h. During this capping process, a significant fraction of the polyamine groups will preferably be anchored in the pores outlets because pore voids are filled with drug. Finally, the obtained solid was filtered under vacuum and then washed with water at pH 2 (adjusted with H_2_SO_4_). The final material MCM-Met-N3 was completely dried under vacuum.

#### 2.2.2. MCM-Saf-N3 Solid Preparation

To obtain this model material, 0.5 g of calcined MCM-41 and 140 mg (0.4 mmol) of safranin O were suspended in 15 mL of acetonitrile and stirred to favor pore loading. After 24 h, an excess of N′-(3-trimethoxysilyl) propyl diethylenetriamine (2.5 mL, 7.5 mmol) was added and the suspension was left under stirring for 6 h. Finally, the obtained solid was filtered under vacuum and washed with water at pH 2. The final MCM-Saf-N3 solid was completely dried under vacuum.

#### 2.2.3. Characterization

Materials characterization was carried out using standard techniques for hybrid organic–inorganic materials, such as powder X-ray diffraction analysis, thermogravimetry and transmission electron microscopy.

##### X-ray Powder Diffraction Analysis

X-ray diffraction patterns of mesoporous silica particles as-made, calcined mesoporous silica, solid MCM-Saf-N3 and solid MCM-Met-N3 were registered using a Bruker AXS D8 Advance diffractometer working at 40 kV/40 mA and using CuKα radiation. All measurements were recorded in the 2θ interval between 0.73° and 10°.

##### Thermogravimetry

Thermograms for the different solid samples were obtained on a TGA/SDTA 851e thermobalance from Mettler Toledo (Mettler Toledo Inc., Schwarzenbach, Switzerland). Samples (4–5 mg) were submitted to a dynamic heating step at 10 °C/min, starting at 25 °C to 1000 °C. Then temperature was maintained at 1000 °C for 30 min. Total organic matter amount was evaluated in the range between 200 and 800 °C. Transmission electron microscopy images were obtained using a JEOL JEM-1010 microscope. Samples were prepared by dropping 10 μL of the corresponding material suspended in water onto carbon-coated copper grids, which were left at room temperature for 24 h until complete water evaporation. Scale bars were included using TEM analysis imaging software.

##### ^1^H NMR Analysis

^1^H NMR spectra were obtained with a Bruker AV400 equipment to evaluate the polyamines amount in each final material. For this, 9 mg of each solid was dissolved in a mixture of 300 μL of NaOD and 700 μL of D_2_O. Then, 4 mg of tetraethylammonium bromide was added as internal standard in each sample. Measurements were performed after 3 h to assure a complete dissolution of the material.

### 2.3. In Vitro Release Assay of Metronidazole from Mesoporous pH-Dependent Molecular Gate Silica Microparticles for Biological Control of Giardia Intestinalis

Release assays were performed to check the functionality of the molecular gate. To perform, in vitro release assays, 5 mg of the solid (MCM-Met-N3 microparticles or MCM-Saf-N3 microparticles) were suspended in 12.5 mL of solutions at different pHs that mimic gastrointestinal standard conditions and are the media recommended by pharmaceutical guides for the evaluation of oral formulations [28,29].

The aqueous buffer solutions at pH 1.2 and 4.5 were prepared using a globally harmonized protocol described by WHO [30]. For pH between 1.2, the solution described in the European Pharmacopoeia which contains sodium chloride and hydrochloric acid with salt concentration of 50 mM was used. (EP) Solution of pH 2 has been obtained with sulfuric acid and solution of pH 4.5 was prepared with sodium acetate, acetic acid, and a salt concentration of 36.5 mM.

However, the pH of the different sections of the gastrointestinal tract depends on many variables such as prandial condition, time after food administration, food volume and content, as well as the volume of secretions. For that reason, in addition to solutions mimicking standard conditions, other buffer solutions described in the literature as simulated gastric fluid in postprandial state (FeSSGF) [31,32] or simulated intestinal fluid in postprandial state (FeSSIF) [33,34] were tested as biorelevant media.

Release profiles were determined by collecting samples at prefixed times (15 min, 30 min, 1 h, 2 h, 3 h, 4 h, 6 h, 8 h, 10 h, and 24 h) filtering them with Teflon filters and determining the amount of dye by fluorescence spectroscopy (λ_exc_: 555 nm, λ_em_: 585 nm) or drug by HPLC.

Weibull, First order and Korsmeyer–Peppas kinetics are the regular kinetic models were fitted to the data (fractions released, *F_rel_*).

Weibull Equation (1):(1)$$F_{rel}=100\cdot (1-e^{\left(-\frac{\left(t^{\beta }\right)}{\alpha }\right)})$$
where *F_rel_* are fractions released and *β* and *α* are the Weibull parameters

First order Equation (2):(2)$$F_{rel}=100\;\cdot \;(1-e^{\left(-kd\cdot t\right)})$$
where *kd* is the rate constant. First order equation is a particular case of Weibull model when *β* = 1; then *kd* = 1/*α*

Korsmeyer–Peppas kinetics, Equation (3):(3)$$F_{rel}=k_{KP}\;\cdot \;\left(t^{n}\right)$$
where *k_KP_* is the rate constant and *n* is Korsmeyer–Peppas parameter.

The best model was selected based on several parameters as the correlation coefficient of experimental versus predicted values, sum squared residual, Akaike’s information criteria (AIC). If models have different number of parameters, the residual variance comparison with Snedecor’s F tests should be calculated to verify that simplex model is not the best model.

### 2.4. Culture of Caco-2 Cells

Human colon adenocarcinoma cells Caco-2 were acquired from the American Type Culture Collection (ATCC, Manassas, VA, USA), and they were grown in DMEM medium supplemented with 10% FBS, 1% penicillin/streptomycin antibiotic and 1% nonessential amino acids. Cells were maintained at 37 °C in incubator, with a humidified controlled atmosphere composed of 5% CO_2_ and 95% air. Passages were carried out when 80% confluence was reached.

#### Caco-2 Cell Monolayer Culture

To obtain Caco-2 monolayers, 2.5 × 10^5^ cells in 2 mL of medium were seeded in the apical side of each insert included in each well of a 6-well plates onto PET porous Millicell hanging cell culture inserts (Merck Millipore; Merck, Schnelldorf, Germany) (area 4.2 cm^2^; pore size 0.4 µm). Then, 3 mL of medium were added in the basolateral compartment. Cells were grown for 21 day in order to allow the right formation of the tight junctions, the cell expression of all the transporters, and the differentiation into enterocytes. Culture medium was changed every 2–3 days. Before and after all the experiments, transepithelial electrical resistance (TEER) of each insert was measured to confirm the correct formation of confluent intestinal monolayers.

### 2.5. In Vitro Cytotoxicity Assay

#### 2.5.1. Culture of *G. intestinalis* Trophozoites

Giardia WB strain trophozoites, purchased from the American Type Culture Collection (ATCC, Manassas, VA, USA), were grown axenically in 30-mL culture tubes containing Modified TYI¬S¬33 Giardia Medium supplemented with 10% bovine serum, penicillin G (100 U/mL) and streptomycin (100 µg/mL), pH 7 at 37 °C.

#### 2.5.2. Viability Assays

The viability assays were performed in order to verify de biological activity of the delivered metronidazole and was performed following three different protocols in 6-well plates.

The first protocol consisted in adding *Giardia intestinalis* (450,000 trophozoites/well) to the wells of 6-wells plates. Parasites were maintained in incubation for 2 h to allow parasite adhesion to the base of the wells. A suspension MCM-Met-N3 microparticles (2 mg/mL) loaded with metronidazole was then added to the wells. At the prefixed times (1, 2, 4, 6, 8, and 20 h) the medium with dead parasite was removed and the living parasites attached to the wells were removed with ice and counted using a Neubauer chamber. Each measure was done in triplicate.

The second procedure consisted in growing monolayers of Caco-2 cells on the base of the plates for 21 days. After that, 400,000 trophozoites/well were added and they were kept in co-incubation for 2 h. Subsequently, the suspension of MCM-Met-N3 microparticles (2 mg/mL) was added. After 24 h, the monolayers were observed with an optical microscope. Caco-2 monolayers with trophozoites but without treatment were used as negative control and Caco-2 monolayers without parasite or microparticle suspension of metronidazole were used as positive control.

The third procedure consisted in growing monolayers of Caco-2 cells on the inserts for 21 days. After that, 400,000 trophozoites/well were added and they were maintained in co-culture for 2 h. Subsequently, the suspension of MCM-Met-N3 microparticles (2 mg/mL) was added. Inserts without trophozoites or microparticles were used as positive control and inserts with parasites but without microparticles were used as negative control. After 24 h the trans-epithelial electrical resistance measurement (TEER) of the monolayers was measured in all the wells in order to check the integrity of the Caco-2 monolayer, as TEER measurements are dependent of the tight junctions’ formation and they are an easy method to determine the viability of the cell monolayer.

### 2.6. In Vivo Biodistribution Profile

In vivo assays were performed to determine the effect of the encapsulation in the pharmacokinetic parameters. They were performed with metronidazole (API) and metronidazole encapsulated in microparticles MCM-Met-N3 solid. The study was conducted according to the guidelines of the Declaration of Helsinki, and approved by the Institutional Review Board (or Ethics Committee) of the Scientific Committee of the Faculty of Pharmacy, Miguel Hernandez University (center code ES0306500001002), and it followed the guidelines described in the EC Directive 86/609, the Council of the Europe Convention ETS 123 and Spanish national laws governing the use of animals in research. (protocol code 2016/VSC/PEA/00092 and date of approval: 27 June 2017).

Male rats were fasted for 4 h. Animals were divided into three groups (A, B and C) of twelve animals each one and formulations were administrated orally. The group A received 1 mL of MCM-Met-N3 microparticles suspension (150 mg/mL), that was prepared previously by mixing the MCM-Met-N3 solid with H_2_SO_4_ pH = 2; the group B received 1 mL of MCM-Saf-N3 microparticles suspension (150 mg/mL), that, in this case, was prepared by mixing the MCM-Saf-N3 solid with H_2_SO_4_ pH = 2; and, the third group C, used as control, received 1 mL of a metronidazole solution (7.5 mg/mL), prepared by mixing metronidazole with H_2_SO_4_ pH = 2.

At the prefixed times (20 min, 45 min, and 2 h) 4 animals of each group were sacrificed and different samples of blood, stomach, duodenum, jejunum, and ileum tissues were taken [35,36]. The tissues were weight, washed with PBS and wiped with a filter paper; then, threefold volume of PBS by the weight of each tissue was added; and finally, they were homogenized and mixed using a glass homogenizer with a Teflon pestle.

### 2.7. Quantitative Analysis of Drug in In Vivo Samples

The tissue homogenates containing a certain amount of safranin or metronidazole were centrifuged at 10,000 rpm for 10 min. After centrifugation of the homogenate or blood, supernatants were diluted with PBS, and protein were removed from samples using acetonitrile (1:3) [37].

Plasma was deproteinized with frozen methanol (1:2), after that samples were centrifugated at 8000 rpm for 10 min. After that, the concentration of metronidazole or safranin in each supernatant sample was added into a new tube and the centrifugation process was repeated. After that concentration of drugs was determined by HPLC analysis.

For the analysis of the digestive samples each group followed a different protocol. In groups A and B (MCM-Met-N3 and MCM-Saf-N3 microparticles) each sample was divided in two tubes. A part of the samples’ supernatant was retired and analyzed directly by HPLC. A second fraction of the supernatants was treated with an acetate solution pH = 4.5 to open artificially the microparticles. 24 h later, lumen samples were centrifuged again, and the supernatant were analyzed by HPLC. In the last group, C (metronidazole solution), samples (supernatants) were directly analyzed by HPLC.

### 2.8. Analysis of Samples

Samples were analyzed by HPLC (Waters 2695) using a Nova-Pak C18 column (4 µM, 3.9 × 150 mm) at 30 °C and a flow-rate of 1.0 mL/min. For metronidazole detection an ultraviolet detector (λ = 248 nm) was used, and the mobile phase was a mixture of water and Acetonitrile (50:50, *v*/*v*), for safranin detection, a fluorescence detector (λ = 520 nm; λ = 585 nm) and the same mobile phase, water, and acetonitrile (50:50, *v/v*), were used. HPLC technique was selected because it offers a high degree of selectivity and specificity. Method was validated previously used. Accuracy was estimated with measuring more than 5 standards, analyzed at least three times, and calculating the associated percentage error (relative error < 10%). Precision or repeatability was calculated as the coefficient of variation of three analysis over the same sample (SD < 5%). Moreover, linearity was established over the range of concentrations present in the samples for every compound (r^2^ > 0.99). The limit of quantitation and detection of both drugs were lower and sample concentration measured. Chromatograms of both compounds are included in Appendix A.

### 2.9. Statistical Analysis

Values are expressed as mean ± standard deviation (SD). To determine statistically significant differences among the experimental groups, groups were evaluated with analysis of variance (ANOVA) and Scheffé post hoc test, and the two-tailed t-Student test for two-group comparison where appropriate. A significance level of 0.05 was selected for all of tests done. The statistical analyses were made with the statistical package SPSS, V.20.00.

## 3. Results and Discussion

### 3.1. Synthesis of Active Materials

In this work, microparticulated mesoporous silica having hexagonally arranged mesopores of ca. 2 nm was selected due to its known high loading capacity, biocompatibility, and ease of functionalization. To confer controlled release properties to the material, polyamines were selected as molecular gates. In polyamine gated mesoporous silica a pH-driven open–close mechanism was previously reported that arises from the hydrogen-bonding interaction between amines at neutral pH (open gate) and Coulombic repulsions at acidic pH between closely located polyammoniums at the pore openings (closed gate). From reported molecular dynamics simulations it was observed that unprotonated amines display poor coverage of the pore (fully open gate), whereas completely protonated amines (simulating a pH 2 or lower) result in a clear reduction of the pore aperture, in agreement with the experimental results. Additionally, to the pH-driven protocol, opening–closing of the gate-like ensemble can also be modulated via an anion-controlled mechanism. The choice of a certain anionic guest results in a different gate-like ensemble behavior, ranging from basically no-action (in the presence of small anions such as chloride) to complete or partial pore blockage, at acidic pH in the presence of sulfate, and phosphate. This anion-controllable response of the gate-like ensemble was explained in terms of anion complex formation with the tethered polyamines.

Following this approach, mesoporous silica microparticles functionalized with a lineal triamine and loaded with the dye safranin O (MCM-Saf-N3) or metronidazole (MCM-Met-N3) were prepared. Both materials were obtained following a similar procedure [38] which consisted on a first loading step, by suspending MCM-41 material in a concentrated solution of the molecule to be encapsulated, followed by the pore capping achieved by functionalization of the external surface with N′-(3-trimethoxysilyl)propyl diethylenetriamine and a final washing with acidic water containing sulfate anions to favor a good pore blockage.

### 3.2. Characterization of Materials

The starting mesoporous silica microparticles (as-made before surfactant removal and after calcination) and solids MCM-Saf-N3 and MCM-Met-N3 were fully characterized by powder X-ray diffraction (PXRD), transmission electron microscopy (TEM), and thermogravimetric studies (Figure 1 and Table 1).

PXRD of the mesostructured material before calcination (as-made) shows reflections indexed as (100), (110), (200), and (210) Bragg peaks which typically observed in MCM-41-like mesoporous materials and indicative of a hexagonal array pore organization. After calcination, the displacement of the (100) peak was appreciated, which can be associated with a significant cell shrinkage caused by the condensation of silanol groups upon calcination. Solids MCM-Saf-N3 and MCM-Met-N3 diffractograms show a similar pattern in which reflections are substantially smoothed due to the lower contrast between the silica walls and the loaded pores. However, the conservation of the (100) diffraction in the final solids confirmed that the mesoporous 3D scaffolding has not been modified or damaged during the loading and functionalization processes.

The mesostructure of the final functionalized solids was also studied by TEM. Figure 2 shows representative TEM images of the different prepared solids. As it can be appreciated, the ordered porous structure observed for starting mesoporous silica was maintained in the final solid loaded with metronidazole and in the material loaded with safranin. Moreover, from low-magnification obtained images, it can be confirmed that the heterogeneous morphology of the initial particles was not modified within the loading and functionalization process to obtain solids MCM-Saf-N3 and MCM-Met-N3.

Organic matter content of solids MCM-Met-N3 and MCM-Saf-N3 was calculated by thermogravimetry and ^1^H-NMR. In the thermogravimetric studies, three regions in which take place different processes depending on the temperature can be differentiated. In the first region (25–180 °C) mass decrease is associated to solvent loss evolution. The second region, from 180 to 800 °C, corresponds to the burning of the organic content present in the material. In our case the joint combustion of the polyamine and the drug or dye encapsulated was observed. In the third region (800–1000 °C) a weight loss was registered due to the loss of water molecules as a result of condensation of silanol groups of the material. In addition, to obtain more information, materials were dissolved in NaOD and ^1^H-NMR studies were performed; the characteristic peaks of the aliphatic chain of the polyamine derivative were identified, integrated, and the amount of polyamines was calculated by means of a calibration curve. Finally, the amount of loaded molecules in each material was calculated combining thermogravimetric analyses and ^1^H-NMR studies. First, total organic content was obtained from the 180–800 °C of the corresponding thermogram. As total organic amount corresponds to the sum of polyamines and loading, safranin or metronidazole content was calculated by subtracting polyamines amount calculated by ^1^H-NMR studies from total organic content. These calculations allowed us to further determine the content of safranin O and metronidazole in the prepared solids (Table 2). The obtained results are similar to those obtained for other gated materials.

### 3.3. In Vitro Release Assays

The absorption of a drug administered in an oral form depends on its release from the formulation, its dissolution at physiological conditions and its permeability through the gastrointestinal tract. All these issues are conditioned by the pH changes typical of an in vivo system. The design of the MCM-Met-N3 and MCM-Saf-N3 materials aims to take advantage of these pH changes to induce cargo release in the area of the intestinal tract in which the parasite *Giardia intestinalis* is concentrated.

In a first step delivery studies at different pHs were performed using the solids MCM-Saf-N3 and MCM-Met-N3. In a typical assay, 5 mg of the corresponding solid was added to 12.5 mL of the selected release media and samples were taken at scheduled times and analyzed by fluorescence for solid MCM-Saf-N3 (Figure 3) or by HPLC for solid MCM-Met-N3 (Figure 4).

First, the gating performance was evaluated following the release of safranine O from solid MCM-Saf-N3 at pH = 1.2 (stomach), pH = 2 (standard acidic condition), and pH = 4.5 (duodenum). Figure 4 shows safranine O release profiles at the mentioned conditions. Kinetics profiles allow to observe that safranine O release is negligible at pH = 1,2 or pH = 2 whereas an important release was observed when the release study was carried out in the buffer at pH = 4.5.

In a second step, release of metronidazole from solid MCM-Met-N3 at pH = 1.2 corresponding to stomach pH was studied. The obtained results indicated that drug release is practically insignificant, even after 24 h. Identical tests were performed in a buffered medium at pH = 4.5 and they revealed a continuous release of metronidazole during several hours (Figure 4).

Taking into account that release occurs at slightly acid pH (pH = 4–5) it is interesting to briefly describe different conditions in duodenum depending on the food intake. In order to mimic the physiological conditions that occur after ingestion of food, it would be appropriate to use a homogenate of the ingested food, because gastric composition in postprandial state depends on the type of food that has been taken. However, at these conditions, it is difficult to perform reproducible experiments to evaluate drug release. Because of that, alternative mediums have been created to simulate the main characteristics of the gastric and intestinal fluids after food intake, taking into account the concentration of each component, the pH in humans and the buffer capacity measured in animal models [39]. For these purpose gastric and intestinal fluids that simulate postprandial state (FeSSGF and FeSSIF) including enzyme and bile salts have been designed and commercialized. In this context, the microparticulated MCM-Met-N3 solid was tested in the presence of the mentioned biorelevant media (FeSSGF and FeSSIF) which simulate the presence of food in the stomach and intestine, respectively. Results (Figure 5) show low release in FeSSGF medium while a slight higher release (near 40% of the cargo) occurs using the media that mimics the presence of food in intestine FeSSIF.

According to this in vitro assay, a sustained release of metronidazole in duodenum could be obtained in absence and presence of food. However, the desirable release is found without food ingestion as, in these conditions, the metronidazole delivery from the microparticles is more effective.

Each release profile has been fitted according to three kinetic models: Weibull, First Order kinetic model and Korsmeyer–Peppas kinetics. These models are selected due to are common model used. The results of of parameters are summarizing in Table 3.

For both molecules (safranine and metronidazole) the comparison of the two simplest models (first order and Korsmeyer–Peppas), which have the same number of parameters, gives as the best dissolution kinetics the first order one, whose sum of squares is lower. On the other hand, the comparison of the first order model and the Weibull model with the Snedecor’s F test, concludes that the best kinetics for describing the release process of safranine is the Weibull one, while for metronidazole, the best one is the first order.

### 3.4. Viability Assays

In a step forward, the performance of solid MCM-Met-N3 in the presence of *Giardia intestinalis* was studied. Viability tests performed on the plates with *Giardia intestinalis* and MCM-Met-N3 microparticles (2 mg/mL), counting with Neubauer chamber, showed that the parasite population had decreased 92.4% after 24 h. Figure 5 shows the parasite destruction curve in the presence of the microparticles at pH 4.5. Results indicate that the metronidazole delivery from the microparticles is able to reduce the population of *Giardia intestinalis.*

In addition, viability assays were performed in “co-cultivation” system of Caco-2/*Giardia* in which g MCM-Met-N3 microparticles suspension (2 mg/mL) were added after Caco-2 monolayers were incubated for 2 h with 400,000 trophozoites/well. The experiments were carried out both on the plates and inserts. Assays were performed using 6-wells plates and three groups: Caco-2 without parasites (positive control), Caco-2 co-cultivated with parasites for 24 h (negative control), and Caco-2 co-cultivated with parasites and with MCM-Met-N3 (test assay). After 24 h, the plates were observed with the microscope (Figure 6) founding that in the wells with microparticles (Figure 6b) any parasite was found, and Caco-2 monolayer showed confluence. On the contrary, in negative control wells (without microparticles (Figure 6c) large damage in Caco-2 monolayers (big holes) were observed.

Similar assays were carried out by growing the Caco-2 monolayer in the inserts. After 24 h positive control TEER measurements in the insert showed values of 262 ± 3 while in negative control inserts, where *Giardia* was incubated 24 h with the Caco-2 monolayers, TEER values indicated that the membrane was damaged (values of 64 ± 6). In the test group, the TEER values were 260 ± 5 which indicates that TEER measurements were adequate and monolayer was not damaged. These findings show that the drug contained in the microparticles was delivered and it was able to perform its anti-parasite function successfully.

### 3.5. In Vivo Assays

In order to verify the metronidazole microparticles advantages in vivo tests were designed and carried out to compare the pharmacokinetic characteristics of a metronidazole solution and a suspension of metronidazole-containing microparticles MCM-Met-N3. Both formulations were orally administrated and, at prefixed times, animals were sacrificed to determine the quantity of the drug in each intestinal segment. Results are represented in Figure 7. Striped bars represent the behavior of the metronidazole solution. It can be observed that, at 2 h post-administration, high plasma level of metronidazole and low level (fourfold less than in plasma) of the drug in the target tissue (duodenum) were obtained. Metronidazole plasma levels after MCM-Met-N3 administration (grey bar) indicates that with the microparticulate formulation the systemic absorption of metronidazole is 12 times lower than that of the free drug. Note that lower plasma levels are related with less adverse effects (compared to free metronidazole). Moreover, the graphic indicates that metronidazole released in duodenum 2 h after the administration of MCM-Met-N3 reaches twofold concentration respect to free metronidazole. In addition, at that time, there is an important quantity of metronidazole still inside the microparticles in lumen (checkered bars) and in tissue (black bars). In jejunum, the level of metronidazole released from the microparticles is lower than in duodenum and only a bit higher than the level obtained after the metronidazole solution administration. However, there is metronidazole still available inside of the microparticles in lumen and in tissues too (checkered and black bars).

In ileum the metronidazole levels are very low using both systems but there are microparticles with metronidazole inside. Microparticles do not cross through the intestinal barrier, so metronidazole has to be released in lumen before crossing the membrane. Results indicate that the microencapsulation of metronidazole in pH-responsive mesoporous silica microparticles is very useful to both reduce the plasma levels and increase the duodenum drug levels. Moreover, it might be possible to optimize the formulation by adhesion or magnetism, to increase the retention of the microparticles in the duodenum to allow the complete metronidazole release from microparticles in the target tissue (duodenum) [35]. Results allow us to suggest that this system could be useful for the local treatment of other intestinal pathologies avoiding systemic effects.

## 4. Conclusions

Local delivery of metronidazole for treatment of giardiasis is preferable over systemic treatment in order to avoid systemic toxic effects while preserving the therapeutic efficacy by providing high concentrations of the drug in duodenum. In this work, we have developed and characterized mesoporous silica particles loaded with metronidazole and capped with a pH-responsive gatekeeper. The effectiveness of the loaded nanoparticles against the *Giardia intestinalis* parasite has been verified and we have assayed and compared plasmatic and duodenal drug levels after a metronidazole solution or the gated mesoporous microparticles of metronidazole administration. In vivo results indicate that it is possible to reduce plasma concentrations and to obtain high metronidazole concentrations in duodenum using the suspension of the metronidazole-loaded microparticles. Mesoporous silica materials capped with pH-responsive groups have revealed its great potential to be used as drug carriers for treatment of intestinal diseases in order to have therapeutic effects with minimum undesirable effects. Starting from this point, the formulation could be improved to increase its residence time in duodenum and pH-responsive gated materials could be further explored for the treatment of other intestinal diseases. Moreover, further studies are being developed using a giardiasis animal disease model in order to evaluate both drug levels in duodenum and its efficacy against the parasite.

## Figures and Tables

**Figure 1 pharmaceutics-13-00094-f001:**
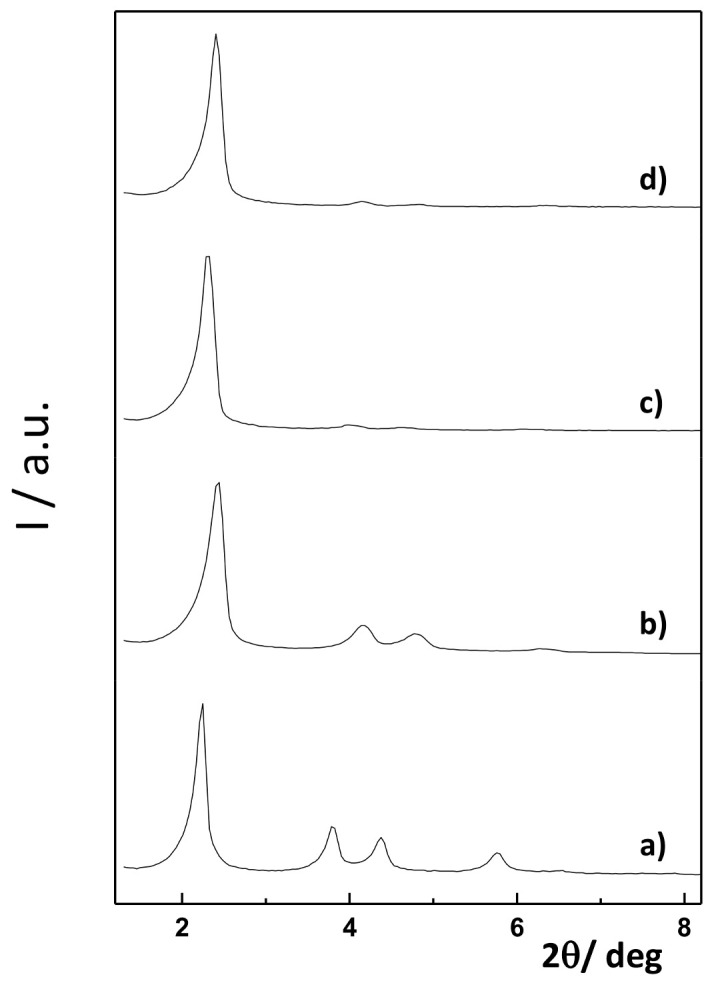
Powder X-ray patterns of (**a**) mesostructured mesoporous silica particles as-made, (**b**) calcined mesoporous silica, (**c**) MCM-Saf-N3, and (**d**) MCM-Met-N3.

**Figure 2 pharmaceutics-13-00094-f002:**
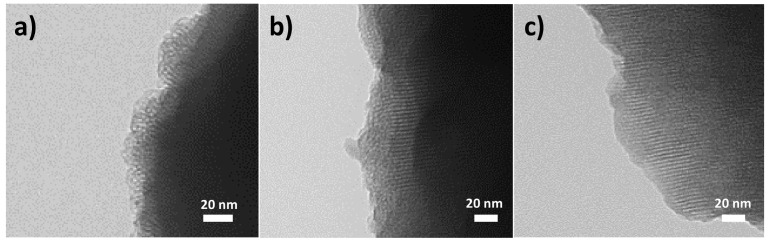
Representative TEM images of (**a**) calcined MCM-41 parent solid, (**b**) MCM-Met-N3 solid, and (**c**) MCM-Saf-N3 solid.

**Figure 3 pharmaceutics-13-00094-f003:**
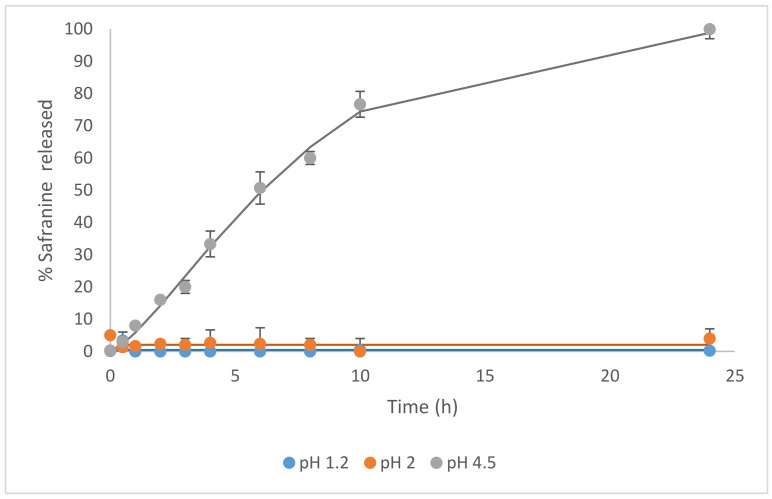
Safranine release profiles from a suspension of mesoporous silica microparticles, functionalized with N3 in the presence of different dissolution media at different pHs.

**Figure 4 pharmaceutics-13-00094-f004:**
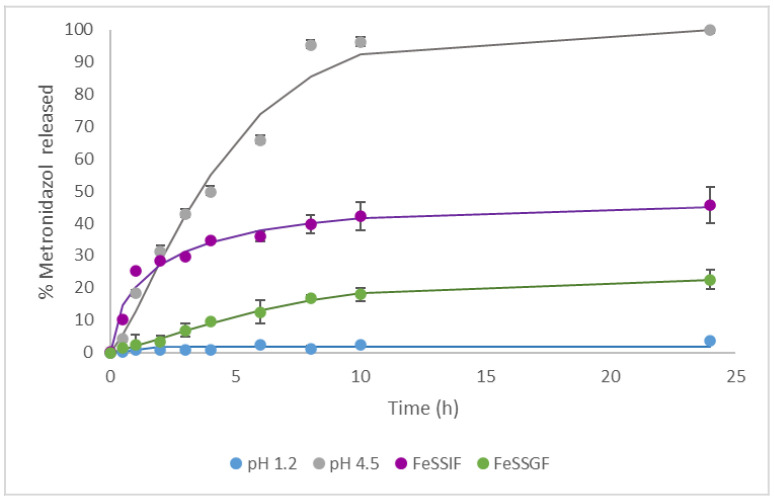
Metronidazole release kinetics from a suspension of mesoporous silica microparticles, functionalized with N3 and loaded with metronidazole (MCM-Met-N3) at different pHs and in the presence of different simulated media.

**Figure 5 pharmaceutics-13-00094-f005:**
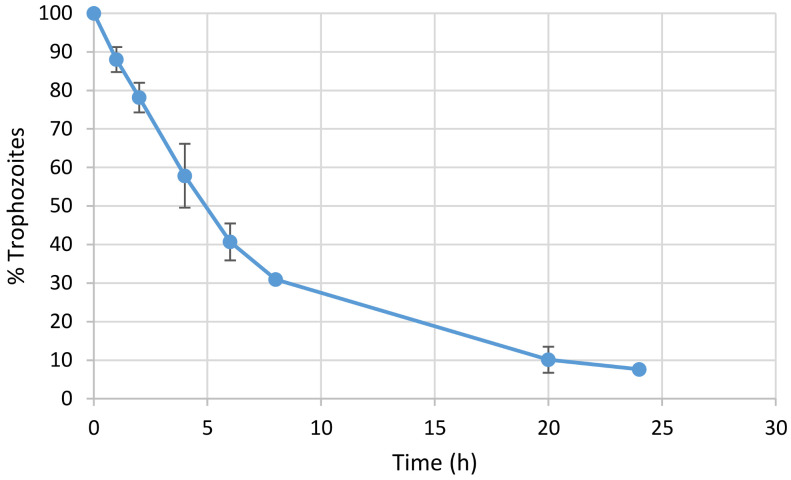
*Giardia intestinalis* destruction curve in the presence of metronidazole microparticles (2 mg/mL) at pH = 4.5.

**Figure 6 pharmaceutics-13-00094-f006:**
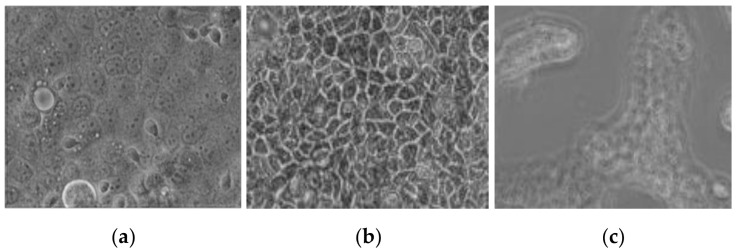
(**a**) Caco-2 monolayer incubated with *Giardia intestinalis* for 2 h without treatment (**b**) Caco-2 monolayer incubated with *Giardia intestinalis* after 24 h of microparticle addition, and (**c**) Caco-2 monolayer incubated with *Giardia intestinalis* after 24 h without treatment. (10× magnification)

**Figure 7 pharmaceutics-13-00094-f007:**
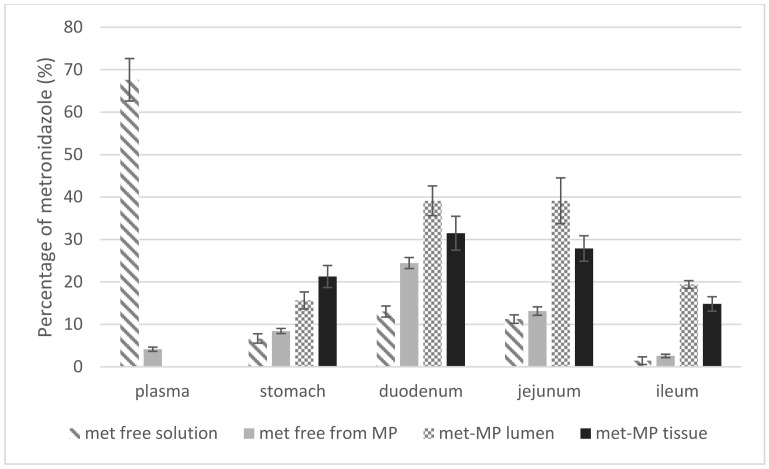
Percentage of metronidazole distributed in plasma and tested tissues 2 h after administration of metronidazole free or metronidazole encapsulated. Metronidazole free (striped bars) or metronidazole encapsulated (grey bars: metronidazole released from microparticles; checkered bars: metronidazole remained inside microparticles in intestinal lumen; black bars: metronidazole inside of microparticles (MP) in tissues).

**Table 1 pharmaceutics-13-00094-t001:** Diffraction reflections of mesostructured mesoporous silica particles as-made, calcined mesoporous silica, MCM-Saf-N3, and MCM-Met-N3.

Solid	2θ (100)	2θ (110)	2θ (200)	2θ (210)
As made MS	2.2	3.8	4.4	5.8
Calcined MS	2.4	4.2	4.8	6.3
MCM-Met-N3	2.4	4.2	4.8	--
MCM-Saf-N3	2.3	4.0	4.6	--

**Table 2 pharmaceutics-13-00094-t002:** Content of safranine O or metronidazole and capping moieties in solids MCM-Met-N3 and MCM-Saf-N3.

Solid	Polyamine (mmol g^−1^)	Safranin (mmol g^−1^)	Metronidazol (mmol g^−1^)
MCM-Met-N3	0.81		0.76
MCM-Saf-N3	0.90	0.36	

**Table 3 pharmaceutics-13-00094-t003:** Summary results of parameters obtained with three kinetics model evaluated.

**First Order Kinetics**				
$F\left(t\right)\;=F_{max}\cdot \;(1-e^{-k_{d}\;\cdot \;t})$				
% Safranine released				
	pH 1.2	pH 2	pH 4.5	
*F_max_* (%)	0.44	2.19	100.00	
*k_d_* (h^−1^)	35.03	1.74	0.11	
% Metronidazole released				
	pH 1.2	pH 4.5	FeSSIF	FeSSGF
*F_max_* (%)	3.98	100.00	40.56	24.43
*k_d_* (h^−1^)	0.08	0.21	0.61	0.12
**Korsmeyer–Peppas kinetics**				
$F\left(t\right)\;=k_{KP}\cdot t^{n}$				
% Safranine released				
	pH 1.2	pH 2	pH 4.5	
*k_KP_* (%·h^−1^)	0.44	1.83	14.44	
*n*	0.00	0.11	0.63	
% Metronidazole released				
	pH 1.2	pH 4.5	FeSSIF	
*k_KP_* (%·h^−1^)	0.51	27.88	22.53	
*n*	0.60	0.45	0.25	
**Weibull kinetics**				
$F\left(t\right)\;=F_{max}\cdot \;(1-e^{-\frac{t^{a}}{b}})$				
% Safranine released				
	pH 1.2	pH 2	pH 4.5	
*F_max_* (%)	0.44	2.04	100.00	
*a*	1.50	3.04	1.36	
*b* (h)	0.00	0.00	16.69	
% Metronidazole released				
	pH 1.2	pH 4.5	FeSSIF	
*F_max_* (%)	1.66	100.00	46.02	
*a*	2.40	1.27	0.60	
*b* (h)	1.49	7.31	1.71	

## Data Availability

Not applicable.

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
