# Peer review of "pH-Dependent Molecular Gate Mesoporous Microparticles for Biological Control of Giardia intestinalis"

_pharmaceutics, 2021, doi:10.3390/pharmaceutics13010094_

Round 1
Reviewer 1 Report
The manuscript (pharmaceutics-1039173) entitled "pH-dependent molecular gate mesoporous microparticles for biological control of Giardia intestinalis" need substantial revision before reconsider for submission in pharmaceutics journal. The authors are advised to consider the following suggestion to improve the quality of manuscript.
- Abstract need to be revised. It should mainly focus on the aim of the current investigation, methodology involved in the current work and their results. The key finding of the work should also be highlighted. The present form of abstract is describing very generalized information.
- Characterization of developed system in methodology section should be divided into different subsection for more elaborative description of method involved.
- In vitro release of the drug should be supported with release kinetics as well.
- Section 2.5 and 2.6 should be merge as "In vitro cytotoxicity assay" and divided into different subsection 2.5.1. "Culture of G. Intestinales Trophozoites", subsection 2.5.2. "Viability Assay", and subsection 2.5.3. "cellular uptake study".
- section 2.7. should be as "in vivo biodistribution profile" and section 2.8. "Quantitative analysis of drug in in vivo sample" should be described with complete detail of chromatogram and validation parameters as supplimentary information.
- Statistical analysis of the obtained data should be carried out to check the level of significance (p value) including SD determination and post hoc analysis.
- In vivo activity of developed formulation system should be included (if possible).
- Results discussion section should required substantial revision with inclusion of characterization results of FT-IR, DSC SEM analysis of developed formulation system. It is suggested to represents the results in table form as well. In present manuscript, representation of results in table for is only 1.
Author Response
Dear reviewer
I have revised the paper following your suggestions. I hope you like it

Reviewer 2 Report
The Authors have submitted a manuscript regarding a nanomaterial-based approach for the oral delivery of metronidazole.
The manuscript is interesting and the claims supported. The topic covered is of interest for the community and the reported results are significant. Overall, this manuscript reports data of enough significance and novelty to warrant publication.
In the following, just few minors the authors should take in consideration:
-fig1: the label has a red highlight that should be removed
-Fig.3&4: error bars should be reported
-some typos as in line 413 "de"
-it's not clear for me why the authors have used cancer cells for their in vitro test instead of non-cancerous endothelial cells (for example bend3 cells)
Author Response
Dear reviewer
I have attached a document with our answer. Thank you very much for you comments

Round 2
Reviewer 1 Report
The revised manuscript (pharmaceutics-1039173) entitled “pH-dependent molecular gate mesoporous microparticles for biological control of Giardia intestinalis” is improved well and authors incorporated the given suggestions nicely. Therefore, in my opinion, the present form of the manuscript should be considered for publication in pharmaceutics journal.